# Influence of Molybdenum and Organic Sources of Copper and Sulfur on the Performance, Carcass Traits, Blood Mineral Concentration, and Ceruloplasmin Activity in Lambs

**DOI:** 10.3390/ani13182945

**Published:** 2023-09-16

**Authors:** Renata Maria Consentino Conti, Thiago Henrique da Silva, Iuli Caetano da Silva Brandão Guimarães, Helena Viel Alves Bezerra, Arlindo Saran Netto, Paulo Henrique Mazza Rodrigues, José Carlos Guilardi Pacheco, Marcus Antonio Zanetti

**Affiliations:** 1Department of Animal Nutrition and Production, College of Veterinary Medicine and Animal Science, University of Sao Paulo, Pirassununga 13635-900, SP, Brazil; renataconti@usp.br (R.M.C.C.); pmazza@usp.br (P.H.M.R.); 2Department of Animal Science, College of Animal Science and Food Engineering, University of Sao Paulo, Pirassununga 13635-900, SP, Brazil; iulivet3@gmail.com (I.C.d.S.B.G.); helena.bezerra@usp.br (H.V.A.B.); saranetto@usp.br (A.S.N.); jospacheco@usp.br (J.C.G.P.); mzanetti@usp.br (M.A.Z.)

**Keywords:** ceruloplasmin activity, mineral antagonism, mineral supplementation, ruminants

## Abstract

**Simple Summary:**

Excessive intake of certain minerals can modify the requirement of others. For example, Mo and S are the main antagonists of Cu, in which high dietary levels of these minerals increase Cu requirements in ruminants. Thus, if Cu levels in the body are low, higher amounts of Mo may become a toxic element for animals. In this paper, we investigated the effects of organic and inorganic sources of Cu and S and their interaction with Mo on the performance, carcass traits, and blood concentration of these minerals, and we also investigated the serum ceruloplasmin activity in lambs. We found that molybdenum and inorganic and organic sources of Cu and S did not improve performance, nor did they have an impact on carcass traits. However, a clear negative effect of Mo and S was detected on Cu bioavailability and metabolism, either due to serum Cu concentrations or through ceruloplasmin activity. However, it was not possible to identify a pattern in the variables studied.

**Abstract:**

This study aimed to evaluate the effects of molybdenum (Mo) and organic and inorganic sources of copper (Cu) and sulfur (S) on the performance, carcass traits, and blood concentration of these minerals in lambs. Forty male non-castrated crossbred Dorper x Santa Inês lambs (20 ± 1.2 kg of body weight and 90 ± 2 d of age) were randomly allocated into one of the ten following treatments: (T0) control, basal diet; (T1) Mo; (T2) inorganic Cu and inorganic S; (T3) inorganic Cu and organic S; (T4) organic Cu and inorganic S; (T5) organic Cu and organic S; (T6) Mo plus inorganic Cu and inorganic S; (T7) Mo plus inorganic Cu and organic S; (T8) Mo plus organic Cu and inorganic S; and (T9) Mo plus organic Cu and organic S. Regardless of the source, Mo, Cu, and S were added at levels of 10 mg, 10 mg, and 2000mg/kg DM, respectively. The mineral supplements (Mo, Cu, and S) were added into the total mixed ration (TMR) by mixing them apart with the mineral and vitamin premix and then put into the TMR. The animals were kept in individual pens and received a total mixed ration for 84 days. Body weight and blood sampling was performed every 28 days. All animals were slaughtered after 84 days, and carcass traits were evaluated. Although organic sources of Cu and S added to Mo supplementation had increased the ADG throughout the study, this effect did not reflect in the heavier final BW outcomes for this treatment. In addition, no effect of these treatments was observed on the carcass traits. The serum Cu concentration was higher for the T0 group compared to the other groups; otherwise, Mo reduced the serum Cu concentration compared to the other groups. Considering the interaction among the minerals and their sources at 84 d of study, organic sources of Cu and S treatment and Mo associated with inorganic sources of Cu plus organic S treatment had an increased serum Cu concentration compared to other groups. Regardless of time, organic sources of Cu and S increased serum S concentration. At 84 days after enrollment, serum Mo concentration was lower for the control group compared to the other groups. Further, Mo supplementation increased its blood concentration compared to the control group throughout the study. The control group had the highest ceruloplasmin activity compared to the other groups; otherwise, at 84 d of the study, either Mo or inorganic S supplementation reduced ceruloplasmin activity. Serum ceruloplasmin activity was higher when Cu supplementation, regardless of source, was associated with organic S. However, at d 84 of the study, inorganic Cu associated to organic S supplements increased serum ceruloplasmin activity. In this current study, it was not possible to identify a pattern in the variables studied, however, further studies are needed to confirm that organic sources of Cu and S interacted alone without a defined pattern.

## 1. Introduction

Supplementation with trace minerals is a husbandry strategy to improve the performance of ruminants and animal health [1,2,3]. However, several negative mineral interactions have been reported in these animals, such as zinc–iron (Zn–Fe), copper–sulfur (Cu–S), copper–molybdenum (Cu–Mo), copper–iron (Cu–Fe), copper–molybdenum–sulfur (Cu–Mo–S) [4,5,6], and copper–sulfur–selenium (Cu–S–Se) [7]. Excessive intake of certain minerals can modify the requirement of others [8]. For example, Mo and S are the main antagonists of Cu, in which high dietary levels of these minerals increase Cu requirements in ruminants [3]. Thus, if Cu levels in the body are low, higher amounts of Mo may become a toxic element for animals [9].

Excessive dietary levels of Mo and S impair Cu absorption through insoluble thiomolybdates and sulfides, which are formed by microbes in the rumen. This can lead to reduced liver Cu concentrations and eventually a reduced blood Cu concentration [10]. Also, thiomolybdate absorption most likely occurs when its production exceeds the amount that can react with Cu to form insoluble complexes in the rumen. This increased thiomolybdate absorption binding to plasma Cu reduces its transport to the tissues [11]. All these mechanisms result in high plasma Cu concentrations, even in deficient animals [3]. Still, this may lead animals to intoxication, promoting red blood cell lysis and, hence, a hemoglobinuric nephrosis state [12].

In addition to the negative impacts of Mo and S on Cu tissue uptake, it has also been reported in sheep that dietary Mo plus S affect another critical step of Cu metabolism—its utilization for ceruloplasmin synthesis in the liver [13]. Ceruloplasmin is a copper–protein complex found in blood responsible for Cu transport, binding up to 95% of this circulating mineral [14]. Consequently, this complex may be an indicator of serum Cu concentration in ruminants [15]. Furthermore, ceruloplasmin is an acute-phase protein that increases during infection or acute stress, being an important substance for animal survivability [14].

There has been a suggestion that providing minerals in an organic form instead of an inorganic form could enhance their bioavailability, leading to increased absorption in the gastrointestinal tract [16]. Having a dietary Cu form that avoids interactions with S and Mo in the rumen, but is still absorbed in the small intestine, is considered advantageous for supplementation [17]. However, the impact of supplying Cu in an organic form on ruminant performance and health remains inconclusive. In addition, Ward et al. [18] proposed that the advantages of providing Cu in an organic form would be more noticeable when dietary concentrations of substances hindering Cu absorption are elevated. Microbes in the rumen metabolize both inorganic and organic sulfur compounds, resulting in the production of sulfide. Additionally, S and Mo combine to create various thiomolybdates (including the mono-, di-, tri-, and tetra-thiomolybdates). These compounds exhibit a robust binding capacity to Cu (particularly the tri- and tetra-thiomolybdates), which bind Cu irreversibly), forming Cu thiomolybdates [6]. However, there is no information, to our knowledge, reporting whether an organic source of S would be detrimental to Cu absorption in ruminants.

Therefore, this study aimed to evaluate the effects of organic and inorganic sources of Cu and S and their interactions with Mo on the performance, carcass traits, and blood concentration of these minerals, as well as to investigate the serum ceruloplasmin activity in lambs.

## 2. Materials and Methods

### 2.1. Animals, Facilities, and Treatments

This study was conducted at the experimental facility of the Department of Animal Science at the College of Animal Science and Food Engineering (FZEA), University of São Paulo (USP), Pirassununga, SP, Brazil. The animals used in this study belonged to the College of Animal Science and Food Engineering, University of São Paulo. All the experimental procedures were reviewed and approved by the Bioethics Committee for Animal Experimentation ay the College of Animal Science and Food Engineering (FZEA/USP; #14.1.1410.74.7).

Forty male non-castrated crossbred Dorper × Santa Inês lambs (20 ± 1.2 kg of body weight and 90 ± 2 d of age, at the start of the experiment) were used in a completely randomized design to evaluate the effects of Mo and organic and inorganic sources of Cu and S on their performance, carcass traits, blood mineral concentrations, and serum ceruloplasmin activity. Throughout the experiment, animals were housed in individual metabolic cages with 1.0 m^2^ of area/animal, under a sheltered barn containing feed bunks and free access to water. The study lasted 84 days, in which 14 days were allowed for facilities and feeding adaptation.

After the adaptation period, the lambs were randomized into the ten following treatments (n = 4 per treatment): (T0) control, basal diet without any additional mineral supplementation; (T1) basal diet added with 10 mg of Mo per kg of dry matter (DM); (T2) basal diet added with inorganic Cu and inorganic S; (T3) basal diet added with inorganic Cu and organic S; (T4) basal diet added with organic Cu and of inorganic S; (T5) basal diet added with organic Cu and organic S; (T6) basal diet added with Mo plus inorganic Cu and inorganic S; (T7) basal diet added with Mo plus inorganic Cu and of organic S; (T8) basal diet added with Mo plus organic Cu and inorganic S; and (T9) basal diet added with Mo plus organic Cu and organic S. Regardless of source, Mo, Cu, and S were added at the levels of 10 mg, 10 mg, and 2000 mg/kg DM, respectively. Treatments comprised the same basal diet, which contained 9.96 mg/kg Cu, 1.02 mg/kg Mo, and 2.56 g/kg S on a DM basis (Table 1). The mineral supplements (Mo, Cu, and S) were added into the total mixed ration (TMR) by mixing them apart with the mineral and vitamin premix and then put into the TMR. Sulfur was supplemented as elemental S (99.0% S; inorganic source) and sulfur proteinate (21.5% S; organic source); copper was supplemented as CuSO_4_ (25.0% Cu; inorganic source) and copper proteinate (11.0% Cu; organic source); molybdenum was supplemented as Na_2_MoO_4_ (39.7% Mo). It is important to highlight that the number of experimental units for the treatments presented herein is in accordance with the study of Zhou et al. [19], who compared coated and uncoated trace elements in growing sheep allocated in metabolic cages.

The basal diet was formulated to meet the nutrient requirements of growing lambs [20]. The animals received a TMR twice a day, at 06:00 h and 17:00 h, in equal amounts in order to maintain refusals at 10%. Ration fed and refusals were weighed daily to control feed intake throughout the experiment. Ration samples of each animal were taken daily during the week of d 28 of each period (the study was composed of 3 periods of 28 d each, for 84 total days period of study) to provide a pooled sample. Immediately after collection, samples were stored at −20 °C until further analysis. The chemical composition presented in Table 1 was assessed in a pooled sample from the control diet (mean values of 3 periods). The Mo, Cu, and S content presented in Table 2 were assessed in a pooled sample of each treatment (mean value of 3 periods).

All samples were dried in a forced air oven at 55 °C for 72 h and ground using 1 mm screen Willey mills (MA340, Marconi, Piracicaba, Brazil) and analyzed for DM (950.15), total N for crude protein estimation (CP; 984.13), and ether extract (EE; 920.39), according to the AOAC [21] methods. Neutral detergent fiber (NDF) and acid detergent fiber (ADF) content were assessed according to Van Soest and Mason [22] using α-amylase without sodium sulfide (TE-149, Tecnal Equipment for Laboratory Inc., Piracicaba, Brazil). Molybdenum, Cu, and S were quantitatively determined using atomic absorption spectrophotometry. Briefly, for Cu determination, a representative 2 g sample was ashed at 500 °C for 5.5 to 6 h. The ash was taken up in 25 mL of 6 *N* hydrochloric acid (HCl), and the solution was aspirated directly into the atomic absorption spectrophotometer. For Mo analysis, a 2 g sample was ashed at 550 °C for 6–8 h. The ash was taken up in 25 mL of 6 *N* HCI, and the solution was aspirated directly into the atomic absorption spectrophotometer. For S determination, samples were weighed into glass tubes, which were digested using nitric acid and perchloric acid at 210 °C. Atomic absorption spectrophotometry was used to determine the concentrations of sulfur by monitoring the plasma emission at 181.975 nm.

### 2.2. Performance, Blood Collection, and Analysis

Body weight (BW) was assessed at enrollment and every 28 d until the end of the study (84 d after enrollment). These measurements were also used to calculate the ADG during the study period [(final weight—initial weight)/period in days].

For blood sampling handling, the animals were restrained in the pens to minimize movement and stress. Blood samples were collected at enrollment (0 d), and 28, 56, and 84 d later through jugular venipuncture using a Vacutainer tube without anticoagulant and a 20-gauge × 2.54-cm Vacutainer needle (Becton, Dickinson and Co, Franklin Lakes, USA). After collection, the tubes were immediately placed in a cooler containing iced water and transported to the laboratory for processing. Samples were analyzed or processed within 2 h after collection. Blood samples collected without anticoagulant were centrifuged at 2000× *g* for 15 min at 4 °C for serum separation (SL 16R Centrifuge, ThermoFisher Scientific Inc., Waltham, MA, USA)

Serum samples were used to determine the trace mineral concentration and ceruloplasmin activity. For serum Mo, Cu, and S concentration, the stored serum samples were thawed and diluted at a 1:1 ratio with distilled water. Afterward, serum Mo, Cu, and S were determined using an atomic absorption spectrophotometer. For ceruloplasmin determination, the methodology proposed by Schosinsky et al. [23] was followed. Briefly, *O*-Dianisidine dihydrochloride was used as the substrate, which was converted into a yellow product for ceruloplasmin and oxygen presence determination. A pinkish stable solution was then produced after adding sulfur acid, terminating this enzymatic assay. Subsequently, the absorbance of this solution was measured at 450 nm on a spectrophotometer reader (Perkin Elmer^TM^ Lambda 35 UV/Vis, Waltham, MA, USA). All these analyses were performed at the Laboratory of Minerals of the College of Animal Science and Food Engineering, University of Sao Paulo (FZEA/USP).

### 2.3. Slaughter and Carcass Evaluation

On day 84 of the study, all animals were weighted and slaughtered at the abattoir of the College of Animal Science and Food Engineering, University of Sao Paulo (Pirassununga, Brazil). All procedures were performed according to the Sanitary and Industrial Inspection Regulation for Animal Origin Products of Humanitarian Slaughter Guidelines, as required by Brazilian law [24]. After slaughter, the carcasses were skinned, eviscerated, washed, and weighed for the determination of hot carcass weights (HCWs). After, the pH at 0h was obtained with a digital pH meter (model HI8314, Hanna Instruments, Ronchi di Villafranca, Italy), measured in the longissimus muscle at the height of the 12th rib equipped using a penetration probe. The carcasses were then stored in a cold room (0–2 °C) for 24 h. After this period, the carcasses were again weighed for the determination of cold carcass weights (CCWs), and the pH at 24 h was measured at the same site.

### 2.4. Statistical Analysis

All statistical analyses were performed using SAS software, version 9.4 (SAS Institute Inc., Cary, NC, USA). The normality of the residuals was verified with the Shapiro–Wilk test using the univariate procedure. The experimental unit was the lamb. The parameters were analyzed considering a 2 × 2 × 2 factorial arrangement (with and without molybdenum, organic and inorganic Cu, and organic and inorganic S, in addition to a basal diet with and without molybdenum).

To evaluate the effect of treatments on the performance, serum Cu, S, and Mo concentration, and ceruloplasmin activity throughout the 4 sampling points (enrollment day, 28, 56, and 84 d later), repeat measures models were fitted using multiple mixed linear models with the MIXED procedure. For these repeated-measure models, a first-order autoregressive covariance structure [(AR(1)] was applied to account appropriately for within-lamb residual correlations among the times evaluated. This variance–covariance structure is indicated for equally spaced data collection and assumes correlation decline as a function of time. The variables treatment, time, and their interaction were forced into all statistical models even in the absence of statistical significance. Carcass traits, which did not include the time factor, were analyzed using the MIXED effect procedure with a model that included the fixed effects of treatment.

The orthogonal contrasts studied were as follows: control vs. others = T0 vs. (T1 + T2 + T3 + T4 + T5 + T6 + T7 + T8 + T9); Mo vs. others = T1 vs. (T0 + T2 + T3 + T4 + T5 + T6 + T7 + T8 + T9); control vs. Mo = T0 vs. T1; Cu source = (T3 + T4) vs. (T5 + T6); S source = (T3 + T5) vs. (T4 + T6); Mo vs. Cu source interaction = (T2 + T4 + T7 + T9) vs. (T3 + T5 + T6 + T8); Mo vs. S source interaction = (T2 + T3 + T8 + T9) vs. (T4 + T5 + T6 + T7); Cu source vs. S source = (T3 + T4 + T7 + T8) vs. (T2 + T5 + T6 + T9); and Mo vs. Cu source vs. S source interaction = (T3 + T4 + T6 + T9) vs. (T2 + T5 + T7 + T8). Means were adjusted using the least square means (LSMeans) procedure of SAS and differences were determined with the *t*-test using the PDIFF command. Statistical significance was declared at *p* ≤ 0.05.

## 3. Results

### 3.1. Animals and Experimental Diet

No health disorder occurred with the lambs throughout the experiment. Experimental diets had an identical nutrient composition, only differing in Mo, Cu, and S content (Table 2). The experimental diet had 57.0 g/kg DM of ashes, 42.8 g/kg DM of EE, 282.0 g/kg DM of NDF, 166.4 g/kg DM of ADF, 157.3 g/kg DM of CP, 11.8g/kg DM of calcium, and 5.8 g/kg DM of phosphorus.

No significant treatment effects were observed on body weight measurements (*p* > 0.05; Table 3); however, the ADG was affected by the Mo, Cu source, and S source interaction (*p* = 0.0283; Table 4 depicted in Figure 1). Also, BW and the ADG were significantly influenced by the time (*p* = 0.0001 and *p* = 0.0014, respectively; Table 3 and Table 4). Mineral supplementation did not significantly affect the carcass traits (*p* > 0.05; Table 5).

### 3.2. Blood Mineral Concentration

The data regarding serum Cu concentration was presented in Table 6. At 56 and 84 days after enrollment, the serum Cu concentration was found to be significantly higher for the control group (T0) compared to the other groups (*p* = 0.0212 and *p* = 0.0088, respectively). On the other hand, the serum Cu concentration was lower for the Mo group (T1) when compared to the other treatments (*p* < 0.0024). In addition, at 84 days of study, the interaction among Mo, Cu, and S, significantly affected the serum Cu concentration (*p* = 0.0022; Figure 2).

Regardless of time, organic sources of the Cu and S interaction significantly increased the serum S concentration (*p* = 0.0355; Table 7). At 84 days after enrollment, the serum Mo concentration was lower for the control group (T0) compared to the other groups (*p* = 0.0021; Table 8); otherwise, the Mo group (T1) increased the serum Mo concentration of the lambs at 28, 56, and 84 days after enrollment (*p* = 0.0001; *p* = 0.0001; and *p* = 0.0001, respectively; Table 8). The Mo group (T1) was also found to have increased the serum Mo concentration at 28 and 84 days after enrollment when compared to the control group (T0; *p* = 0.0182 and *p* = 0.0054, respectively; Table 8).

### 3.3. Ceruloplasmin Activity

The control diet (T0) had the significantly highest value of serum ceruloplasmin activity compared to the other treatments (*p* = 0.0053; Table 9). At 84 days after enrollment, ceruloplasmin displayed reduced activity when Mo (*p* = 0.0254) and inorganic S (*p* = 0.0310) were supplemented to the lambs. There was a significant interaction between the Cu and S sources (*p* = 0.0055), in which the serum level of ceruloplasmin was higher when Cu supplementation, regardless of source, was associated with organic S. In addition, there was significant interaction among Mo × Cu source × S source (*p* = 0.0074; Figure 3).

## 4. Discussion

### 4.1. Animals and Experimental Diet

No health problems were diagnosed across all animals during the entirety of the experimental period. This ensures that all the effects detected herein were exclusively through mineral addition into diets. Furthermore, as shown, only mineral contents were different among the treatments. Thus, there was no difference in the experimental basal diet, which may have interfered with the results presented herein. In addition, as a limitation of this study, the experimental unit in each group was low; therefore, there was a high chance of increased type II error occurrence. However, the authors of [19] used a similar design for their study, comparing coated and uncoated trace elements on growth performance, apparent digestibility, intestinal development, and microbial diversity in growing sheep.

### 4.2. Performance and Carcass Traits

The time influenced the ADG, which ranged from 140 to 180 g/day; this was expected as we were studying growing lambs. When contrasts were applied, the inclusion of organic Cu and S added with Mo increased the ADG (0.180 kg/d), when compared to the same sources without Mo addition (0.150 kg/d; Figure 1). Molybdenum is an essential trace mineral required for animal growth; however, its excessive dietary inclusion may impair Cu absorption. Further, dietary S may reduce Cu absorption [25]. Insoluble thiomolybdates and sulfides formed by ruminal microbes bind dietary Cu, decreasing its bioavailability [5,6,8]. Dick et al. [26] reported that the limiting effect of Mo on Cu nutrition is dependent on the level of S in the feed; further, Suttle [27] reported that organic Mo may be a more effective Cu antagonist than its inorganic counterpart. Thus, Mo, Cu, and S supply balancing and sources (inorganic and organic) may affect ruminant growth [16,17]. It is of great importance for lambs, which are more susceptible to Cu toxicity than either cattle or goats [25]. Dezfoulian et al. [28] studied the impacts of doses (of 10 and 20 mg/kg DM) and Cu sources (CuSO_4_ and Cu proteinate) in sheep diets and did not observe any effect on the ADG; however, Cu supplementation, regardless of source, had a significant effect on the feed conversion ratio. Therefore, our findings suggest that the organic–mineral interaction can influence the performance of growing lambs being less sensitive to impaired interactions.

Although an effect was detected regarding the ADG, it did not reflect on heavier body weights in the organic Cu and S added with Mo group. Likewise, the HCW and CCWs also did not differ, ranging from 16.7 to 19.65 kg. These values were similar to those observed by Garrine et al. [29], who supplemented crossbreed Merino × Texel lambs with different doses and sources of Cu. In this study, no influence of treatments was observed on the pH values, which showed normal values for the initial (0 h) and final (24 h) measurements, indicating a normal rate of muscle pH fall due to postmortem glycolysis [30].

### 4.3. Blood Mineral Concentration

The serum concentration of Cu was higher in the control group compared to the other treatments; otherwise, serum Cu levels decreased when supplementing Mo and through the Mo × Cu × S interaction. As previously mentioned, excessive dietary Mo and S may impair Cu absorption through insoluble thiomolybdates and sulfides, formed by ruminal microbes [5,6,10]. This can lead to reduced Cu levels in the liver and eventually reduced plasma Cu concentrations as a consequence [3]. Lower serum Cu concentrations were detected, in this present study, in the Mo-supplemented group compared to the other groups, confirming the negative effect of Mo on Cu absorption in bloodstream. Corroborating this, Dias et al. [25], in their meta-analytical study, found reduced plasmatic Cu concentrations as increasing levels of Mo and S were supplemented into the diet of growing-finishing cattle. Further, Suttle [31] observed reduced serum Cu concentrations by increasing S levels in sheep diets, both in organic and inorganic form. A similar reduction was found in sheep when including high S levels, which increased the concentration of sulfides in ruminants [32]. In this study, all treatments were composed of either Mo or S inclusion, except for the control group, which encompassed, in this study, a higher serum Cu concentration. Thus, Cu absorption, even due to its higher level of supplementation, may be jeopardized by the Mo and S interaction, reflecting in lower serum Cu concentrations.

Otherwise, at 84 days after enrollment, higher serum Cu concentrations were detected in animals receiving organic Cu plus organic S and in animals receiving Mo + inorganic Cu + organic S compared to other treatments. One possible explanation is that thiomolybdate absorption most likely occurs when the production of thiomolybdates exceeds the amount that can react with Cu to form insoluble complexes in the rumen. One of the systemic effects is thiomolybdates binding strongly to albumin-bound plasma Cu, which results in a reduced transport of available Cu for biochemical processes [11]. This can result in high plasma Cu concentrations even in ruminants deficient in Cu [3]. Of note, total serum Cu is of little significance in Mo and S presence, as even though the concentration of Cu may be increased in the plasma, it is not available for tissue absorption due to its insoluble bonds.

In this study, lambs fed with organic Cu and S sources exhibited increased serum S concentrations compared to inorganic sources or the interaction between inorganic and organic sources. This could be attributed to the strong bond between Cu and S in the bloodstream, which increases serum S retention by decreasing its tissue uptake [11].

The higher serum Mo concentrations found in this study for Mo-fed animals are in agreement with those found by Pott et al. [33] and Galbraith et al. [34], who observed an increase in plasma Mo concentration when 7.5 mg Mo/kg DM or 15 mg Mo/kg DM were added to the sheep and goat diets, respectively. Interestingly, the control diet had higher serum Mo levels than the treatments that received Cu and S supplementation with no Mo addition, regardless of sources. This may be through the negative effect of S, which has been reported to reduce the plasma concentration of Mo, regardless of its source [27]. Overall, organic S may be the best strategy to improve Cu bioavailability in the blood of lambs when inorganic Cu and Mo are supplemented; otherwise, when Mo is not supplemented and organic S is supplemented, organic sources of Cu are recommended.

### 4.4. Ceruloplasmin Activity

Regarding serum ceruloplasmin activity, the reduced values found in the presence of Mo, inorganic S, or their interaction all corroborate with McDowell [35], who indicated that lower Cu availability occurs as a result of a lower level of ceruloplasmin activity. It is due to the copper –molybdenum–sulfur interaction, that the increase of Mo and S in the diet, alone, or in combination, can reduce Cu use [3,5,6,10,26,32], as it reduces the Cu absorption [11,33]. Overall, organic sources of S associated with inorganic or organic Cu with or without Mo may be the best strategy to increase its activity.

## 5. Conclusions

Molybdenum and inorganic and organic sources of Cu and S did not improve performance, nor did they have an impact on carcass traits in this study. However, a clear negative effect of Mo and S was detected on Cu bioavailability and metabolism, either by serum Cu concentrations or by ceruloplasmin activity. In this study, it was not possible to identify a pattern in the variables studied; however, further studies are needed to confirm that organic sources of Cu and S interacted alone without a defined pattern.

## Figures and Tables

**Figure 1 animals-13-02945-f001:**
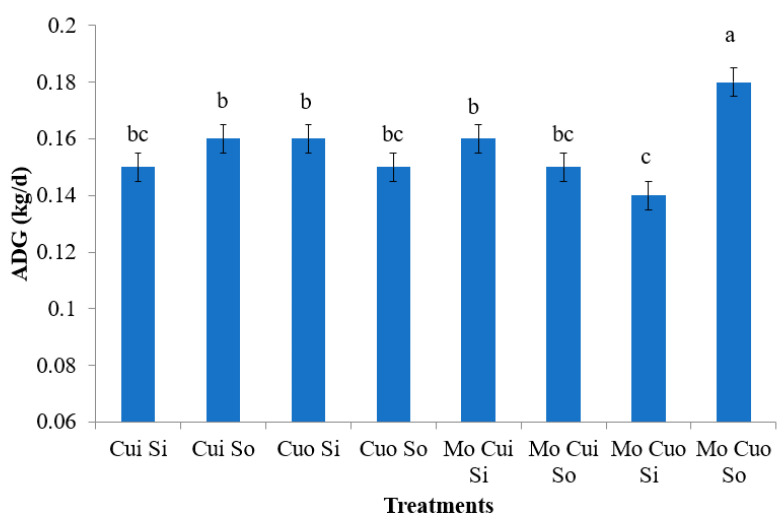
Average daily gain (ADG) of lambs supplemented with molybdenum (Mo) and inorganic (i) or organic (o) sources of copper (Cu) and sulfur (S) during 84 d. ^a–c^ Means with different superscripts significantly differ, as evidenced with the *t*-test (*p* < 0.05). Bars represent the pooled SEM.

**Figure 2 animals-13-02945-f002:**
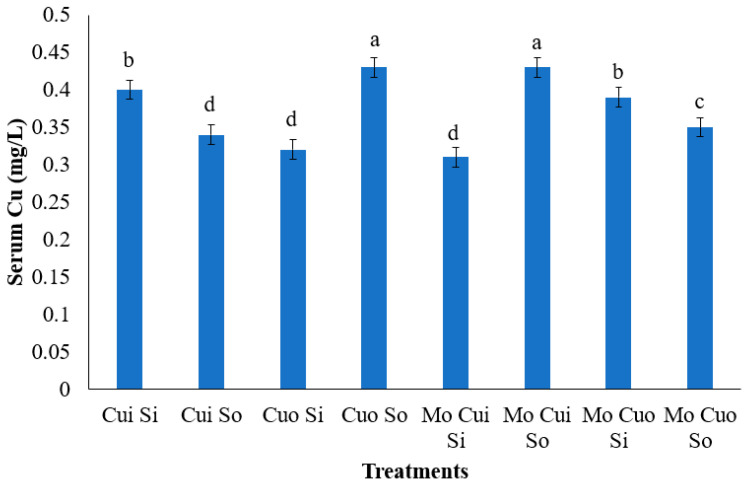
Serum copper (Cu) concentration at d 84 of lambs supplemented with molybdenum (Mo) and inorganic (i) or organic (o) sources of copper (Cu) and sulfur (S). ^a–d^ Means with different superscripts significantly differ, as evidenced with the *t*-test (*p* < 0.05). Bars represent the pooled SEM.

**Figure 3 animals-13-02945-f003:**
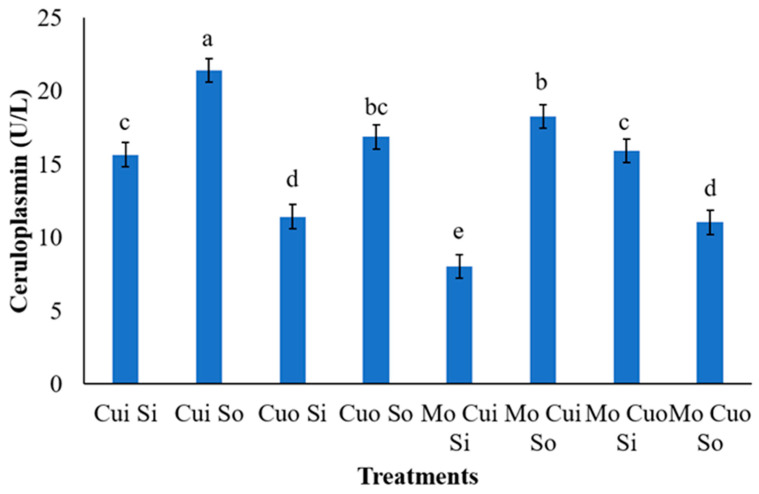
Serum ceruloplasmin activity at d 84 of lambs supplemented with molybdenum (Mo) and inorganic (i) or organic (o) sources of copper (Cu) and sulfur (S). ^a–e^ Means with different superscripts significantly differ, as evidenced with the *t*-test (*p* < 0.05). Bars represent the pooled SEM.

**Table 1 animals-13-02945-t001:** Ingredients and chemical composition of the basal diet (g/kg DM, otherwise stated).

Item		Basal Diet
Ingredients		
	Corn meal	553.0
	Wheat middlings	10.0
	Cottonseed hulls	250.0
	Soybean meal	125.0
	Soybean oil	10.0
	Limestone	12.0
	Dicalcium phosphate	20.0
	Urea	10.0
	Vitamin premix ^a^	05.0
	Mineral premix ^b^	05.0
Chemical composition		
	Dry matter (g/kg OM ^c^)	906.5
	aNDF	282.0
	aADF	166.4
	Crude protein	157.3
	Ether extract	42.8
	Ash	57.0
	Calcium	11.8
	Phosphorus	5.8

^a^ Containing (per kg): 400,000 U of vitamin A and 4000 UI of vitamin E. ^b^ Containing (per kg): 100 mg of I, 4000 mg of Fe, 40 mg of Co, 3000 mg of Mg, 40 mg of Se, 4000 mg of Zn, and 216 g of NaCl. ^c^ Organic matter.

**Table 2 animals-13-02945-t002:** Copper, molybdenum, and sulfur contents in treatments (mg/kg DM).

Treatment	Molybdenum	Copper	Sulfur
T0	1.02	9.96	2561.88
T1	9.54	8.42	2369.52
T2	1.05	15.70	4340.04
T3	0.97	15.72	5178.19
T4	1.18	17.22	4829.88
T5	0.90	17.61	5209.69
T6	11.60	15.74	4645.46
T7	10.21	19.91	5288.51
T8	11.92	16.67	5190.82
T9	11.72	17.16	4906.29

(T0) Control, basal diet without any additional mineral supplementation; (T1) basal diet added with 10 mg of Mo; (T2) basal diet added with 10 mg of inorganic Cu per kg DM and 0.2% of inorganic S; (T3) basal diet added with 10 mg of inorganic Cu and 0.2% organic S; (T4) basal diet added with 10 mg of organic Cu and 0.2% of inorganic S; (T5) basal diet added with 10 mg of organic Cu and 0.2% of organic S; (T6) basal diet added with 10 mg of Mo plus 10 mg of inorganic Cu and 0.2% of inorganic S; (T7) basal diet added with 10 mg of Mo plus 10 mg of inorganic Cu and 0.2% of organic S; (T8) basal diet added with 10 mg of Mo plus 10 mg of organic Cu and 0.2% of inorganic S; and (T9) basal diet added with 10 mg of Mo plus 10 mg of organic Cu and 0.2% of organic S. Performance and carcass traits.

**Table 3 animals-13-02945-t003:** Body weight (kg) of lambs receiving a control diet or supplemented with molybdenum (Mo) or sources of copper (Cu) and sulfur (S).

Mineral Inclusion				Time (Days)				
Mo	Cu	S	Treatment	0	28	56	84	Mean
Without	Without	Without	T0	23.87	27.71	32.91	37.21	30.42
With	Without	Without	T1	21.53	24.31	29.54	34.28	27.41
Without	Inorganic	Inorganic	T2	21.98	24.68	30.28	33.85	27.69
		Organic	T3	22.38	25.65	31.89	35.85	28.94
	Organic	Inorganic	T4	22.61	25.83	31.23	36.01	28.92
		Organic	T5	22.64	25.88	30.94	35.25	28.68
With	Inorganic	Inorganic	T6	22.31	25.63	30.56	35.4	28.48
		Organic	T7	21.66	25.45	30.11	34.8	28.01
	Organic	Inorganic	T8	20.91	23.98	28.79	32.86	26.63
		Organic	T9	21.6	25.26	31.45	36.75	28.77
Principal effects								
Without				22.4	25.51	31.08	35.24	28.56
With				21.62	25.08	30.23	34.95	27.97
	Inorganic			22.08	25.35	30.71	34.98	28.28
	Organic			21.94	25.23	30.6	35.22	28.25
		Inorganic		21.95	25.03	30.21	34.53	27.93
		Organic		22.07	25.56	31.1	35.67	28.6
Average data								
Mean				22.23	25.34	30.87	35.32	28.49
SEM				0.67	0.83	0.96	1.09	0.59
Statistical probabilities of contrasts								
Control vs. others [(T0) vs. (T1 + T2 + T3 + T4 + T5 + T6 + T7 + T8 + T9)]				NS				
Mo vs. others (T1 vs. T0 + T2 + T3 + T4 + T5 + T6 + T7 + T8 + T9)				NS				
Mo presence (T0 vs. T1)				NS				
Cu source [(T3 + T4) vs. (T5 + T6)]				NS				
S source [(T3 vs. T5) vs. (T4 + T6)]				NS				
Mo × Cu [(T2 + T4 + T7 + T9) vs. (T3 + T5 + T6 + T8)]				NS				
Mo × S [(T2 + T3 + T8 + T9) vs. (T4 + T5 + T6 + T7)]				NS				
Cu × S [(T3 + T4 + T7 + T8) vs. (T2 + T5 + T6 + T9)]				NS				
Mo × Cu × S [(T3 + T4 + T6 + T9) vs. (T2 + T5 + T7 + T8)]				NS				

*p*-value for treatment: 0.9986; *p*-value for time: 0.0001. Treatment × time: 0.7909. SEM—standard error of the mean; NS—not significant.

**Table 4 animals-13-02945-t004:** Average daily gain (ADG; kg/d) of lambs receiving a control diet or supplemented with molybdenum (Mo) or sources of copper (Cu) and sulfur (S).

Mineral Inclusion				Time (Days)				
Mo	Cu	S	Treatment	0	28	56	84	Mean
Without	Without	Without	T0		0.14	0.19	0.15	0.16
With	Without	Without	T1		0.1	0.19	0.17	0.15
Without	Inorganic	Inorganic	T2		0.1	0.2	0.14	0.15
		Organic	T3		0.12	0.22	0.14	0.16
	Organic	Inorganic	T4		0.11	0.19	0.17	0.16
		Organic	T5		0.12	0.18	0.15	0.15
With	Inorganic	Inorganic	T6		0.12	0.18	0.17	0.16
		Organic	T7		0.12	0.17	0.17	0.15
	Organic	Inorganic	T8		0.1	0.17	0.15	0.14
		Organic	T9		0.13	0.22	0.19	0.18
Principal effects								
Without					0.11	0.2	0.15	0.15
With					0.12	0.18	0.17	0.16
	Inorganic				0.11	0.19	0.16	0.15
	Organic				0.11	0.19	0.16	0.16
		Inorganic			0.11	0.19	0.16	0.15
		Organic			0.12	0.2	0.16	0.16
Average data								
Mean					0.12	0.19	0.16	0.16
SEM					0.007	0.007	0.007	0.005
Statistical probabilities of contrasts								
Control vs. others [(T0) vs. (T1 + T2 + T3 + T4 + T5 + T6 + T7 + T8 + T9)]				NS				
Mo vs. others (T1 vs. T0 + T2 + T3 + T4 + T5 + T6 + T7 + T8 + T9)				NS				
Mo presence (T0 vs. T1)				NS				
Cu source [(T3 + T4) vs. (T5 + T6)]				NS				
S source [(T3 vs. T5) vs. (T4 + T6)]				NS				
Mo × Cu [(T2 + T4 + T7 + T9) vs. (T3 + T5 + T6 + T8)]				NS				
Mo × S [(T2 + T3 + T8 + T9) vs. (T4 + T5 + T6 + T7)]				NS				
Cu × S [(T3 + T4 + T7 + T8) vs. (T2 + T5 + T6 + T9)]				NS				
Mo × Cu × S [(T3 + T4 + T6 + T9) vs. (T2 + T5 + T7 + T8)]				0.0283				

*p*-value for treatment: 0.2912; *p*-value for time: 0.0014. Treatment × time: 0.7918. SEM—standard error of the mean; NS—not significant.

**Table 5 animals-13-02945-t005:** Carcass traits of lambs receiving a control diet or supplemented with molybdenum (Mo) or sources of copper (Cu) and sulfur (S).

Mineral Inclusion				Variable			
Mo	Cu	S	Treatment	pH 0 h	pH 24 h	Hot Carcass Weight (kg)	Cold Carcass Weight (kg)
Without	Without	Without	T0	6.17	5.62	19.65	19.04
With	Without	Without	T1	6.15	5.63	17.5	16.8
Without	Inorganic	Inorganic	T2	6.48	5.77	17.3	16.7
		Organic	T3	6.46	5.6	18.5	17.9
	Organic	Inorganic	T4	6.36	5.68	19.03	18.33
		Organic	T5	6.16	5.66	19.05	18.43
With	Inorganic	Inorganic	T6	6.6	5.71	18.15	17.5
		Organic	T7	6.45	5.85	18	17.43
	Organic	Inorganic	T8	6.51	5.7	16.8	16.18
		Organic	T9	6.34	5.64	18.93	18.23
Principal effects							
Without				6.36	5.68	18.47	17.84
With				6.47	5.72	17.97	17.33
	Inorganic			6.5	5.73	17.99	17.38
	Organic			6.34	5.67	18.45	17.79
		Inorganic		6.49	5.72	18.82	17.18
		Organic		6.35	5.69	18.62	17.99
Average data							
Mean				6.36	5.68	18.35	17.72
SEM				0.052	0.024	0.608	0.598
Statistical probabilities of contrasts							
Control vs. others [(T0) vs. (T1 + T2 + T3 + T4 + T5 + T6 + T7 + T8 + T9)]				NS	NS	NS	NS
Mo vs. others (T1 vs. T0 + T2 + T3 + T4 + T5 + T6 + T7 + T8 + T9)				NS	NS	NS	NS
Mo presence (T0 vs. T1)				NS	NS	NS	NS
Cu source [(T3 + T4) vs. (T5 + T6)]				NS	NS	NS	NS
S source [(T3 vs. T5) vs. (T4 + T6)]				NS	NS	NS	NS
Mo × Cu [(T2 + T4 + T7 + T9) vs. (T3 + T5 + T6 + T8)]				NS	NS	NS	NS
Mo × S [(T2 + T3 + T8 + T9) vs. (T4 + T5 + T6 + T7)]				NS	NS	NS	NS
Cu × S [(T3 + T4 + T7 + T8) vs. (T2 + T5 + T6 + T9)]				NS	NS	NS	NS
Mo × Cu × S [(T3 + T4 + T6 + T9) vs. (T2 + T5 + T7 + T8)]				NS	NS	NS	NS

SEM—standard error of the mean; NS—not significant.

**Table 6 animals-13-02945-t006:** Serum copper concentration (mg/L) of lambs receiving a control diet or supplemented with molybdenum (Mo) or sources of copper (Cu) and sulfur (S).

Mineral Inclusion			Treatment	Time (Days)				
Mo	Cu	S		0	28	56	84	Mean
Without	Without	Without	T0	0.45	0.43	0.46	0.45	0.45
With	Without	Without	T1	0.35	0.36	0.35	0.25	0.33
Without	Inorganic	Inorganic	T2	0.43	0.39	0.37	0.4	0.4
		Organic	T3	0.44	0.33	0.3	0.34	0.35
	Organic	Inorganic	T4	0.44	0.33	0.3	0.32	0.35
		Organic	T5	0.35	0.4	0.38	0.43	0.39
With	Inorganic	Inorganic	T6	0.3	0.33	0.32	0.31	0.31
		Organic	T7	0.45	0.36	0.41	0.43	0.41
	Organic	Inorganic	T8	0.41	0.36	0.37	0.39	0.39
		Organic	T9	0.3	0.35	0.34	0.35	0.33
Principal effects								
Without					0.36	0.34	0.37	0.37
With					0.35	0.36	0.37	0.36
	Inorganic				0.35	0.35	0.37	0.37
	Organic				0.36	0.35	0.37	0.36
		Inorganic			0.35	0.34	0.36	0.36
		Organic			0.36	0.36	0.39	0.37
Average data								
Mean				0.4	0.37	0.36	0.37	0.37
SEM				0.015	0.013	0.016	0.013	0.007
Statistical probabilities of contrasts								
Control vs. others [(T0) vs. (T1 + T2 + T3 + T4 + T5 + T6 + T7 + T8 + T9)]				NS	NS	0.0212	0.0088	NS
Mo vs. others (T1 vs. T0 + T2 + T3 + T4 + T5 + T6 + T7 + T8 + T9)				NS	NS	NS	0.0024	NS
Mo presence (T0 vs. T1)				NS	NS	NS	NS	NS
Cu source [(T3 + T4) vs. (T5 + T6)]				NS	NS	NS	NS	NS
S source [(T3 vs. T5) vs. (T4 + T6)]				NS	NS	NS	NS	NS
Mo × Cu [(T2 + T4 + T7 + T9) vs. (T3 + T5 + T6 + T8)]				NS	NS	NS	NS	NS
Mo × S [(T2 + T3 + T8 + T9) vs. (T4 + T5 + T6 + T7)]				NS	NS	NS	NS	NS
Cu × S [(T3 + T4 + T7 + T8) vs. (T2 + T5 + T6 + T9)]				NS	NS	NS	NS	NS
Mo × Cu × S [(T3 + T4 + T6 + T9) vs. (T2 + T5 + T7 + T8)]				NS	NS	NS	0.0022	NS

*p*-value for treatment: 0.2413; *p*-value for time: 0.0057. Treatment × time: 0.0001. SEM—standard error of the mean; NS—not significant.

**Table 7 animals-13-02945-t007:** Serum S concentration (mg/L) of lambs receiving a control diet or supplemented with molybdenum (Mo) or sources of copper (Cu) and sulfur (S).

Mineral Inclusion								
Mo	Cu	S						
Without	Without	Without	T0	1227.11	1021.41	1159.69	1170.3	1144.63
With	Without	Without	T1	1437.58	1245.49	1559.48	1479.99	1430.63
Without	Inorganic	Inorganic	T2	1318.22	1144.77	1488.13	1481.75	1358.22
		Organic	T3	1324.11	1121.26	1330.45	1351.19	1281.75
	Organic	Inorganic	T4	1311.83	1127.96	1200.98	1220.72	1215.37
		Organic	T5	1401.44	1255.69	1503.38	1687.64	1462.03
With	Inorganic	Inorganic	T6	1400.63	1224.19	1490.01	1506.15	1405.24
		Organic	T7	1348.67	1200.14	1422.88	1458.74	1357.6
	Organic	Inorganic	T8	1241.01	1021.26	1101.2	1067.52	1107.75
		Organic	T9	1393.11	1234.41	1451.48	1517.24	1399.06
Principal effects								
Without					1162.42	1380.74	1435.32	1329.34
With					1170	1366.39	1387.41	1317.41
	Inorganic				1172.59	1432.87	1449.46	1350.7
	Organic				1159.83	1314.26	1373.28	1296.05
		Inorganic			1129.54	1320.08	1319.04	1271.64
		Organic			1202.87	1427.05	1503.7	1375.11
Average data								
Mean				1334.98	1153.07	1360.72	1383.46	1308.06
SEM				23.36	20.67	49.25	53.39	20.81
Statistical probabilities of contrasts								
Control vs. others [(T0) vs. (T1 + T2 + T3 + T4 + T5 + T6 + T7 + T8 + T9)]				NS	NS	NS	NS	NS
Mo vs. others (T1 vs. T0 + T2 + T3 + T4 + T5 + T6 + T7 + T8 + T9)				NS	NS	NS	NS	NS
Mo presence (T0 vs. T1)				NS	NS	NS	NS	NS
Cu source [(T3 + T4) vs. (T5 + T6)]				NS	NS	NS	NS	NS
S source [(T3 vs. T5) vs. (T4 + T6)]				NS	NS	NS	NS	NS
Mo × Cu [(T2 + T4 + T7 + T9) vs. (T3 + T5 + T6 + T8)]				NS	NS	NS	NS	NS
Mo × S [(T2 + T3 + T8 + T9) vs. (T4 + T5 + T6 + T7)]				NS	NS	NS	NS	NS
Cu × S [(T3 + T4 + T7 + T8) vs. (T2 + T5 + T6 + T9)]				NS	NS	NS	NS	0.0355
Mo × Cu × S [(T3 + T4 + T6 + T9) vs. (T2 + T5 + T7 + T8)]				NS	NS	NS	NS	NS

*p*-value for treatment: 0.1992; *p*-value for time: 0.0001. Treatment × time: 0.4175. SEM—standard error of the mean; NS—not significant.

**Table 8 animals-13-02945-t008:** Serum molybdenum concentration (mg/L) of lambs receiving a control diet or supplemented with molybdenum (Mo) or sources of copper (Cu) and sulfur (S).

Mineral Inclusion			Treatment	Time (days)				
Mo	Cu	S		0	28	56	84	Mean
Without	Without	Without	T0	0.036	0.014	0.055	0.054	0.07
With	Without	Without	T1	0.022	0.89	0.715	0.991	0.654
Without	Inorganic	Inorganic	T2	0.032	0.075	0.212	0.032	0.087
		Organic	T3	0.028	0.072	0.027	0.027	0.039
	Organic	Inorganic	T4	0.024	0.068	0.031	0.027	0.037
		Organic	T5	0.027	0.087	0.024	0.026	0.041
With	Inorganic	Inorganic	T6	0.029	0.141	0.097	0.113	0.095
		Organic	T7	0.033	0.142	0.128	0.103	0.102
	Organic	Inorganic	T8	0.042	0.146	0.147	0.139	0.118
		Organic	T9	0.027	0.134	0.064	0.097	0.081
Principal effects								
Without					0.075	0.073	0.028	0.051
With					0.141	0.109	0.113	0.099
	Inorganic				0.108	0.116	0.069	0.081
	Organic				0.109	0.067	0.072	0.069
		Inorganic			0.107	0.122	0.078	0.085
		Organic			0.109	0.06	0.063	0.066
Average data								
Mean				0.03	0.187	0.145	0.156	0.129
SEM				0.0029	0.0372	0.0372	0.0441	0.0176
Statistical probabilities of contrasts								
Control vs. others [(T0) vs. (T1 + T2 + T3 + T4 + T5 + T6 + T7 + T8 + T9)]				NS	NS	NS	0.0021	NS
Mo vs. others (T1 vs. T0 + T2 + T3 + T4 + T5 + T6 + T7 + T8 + T9)				NS	0.0001	0.0001	0.0001	NS
Mo presence (T0 vs. T1)				NS	0.0182	NS	0.0054	NS
Cu source [(T3 + T4) vs. (T5 + T6)]				NS	NS	NS	NS	NS
S source [(T3 vs. T5) vs. (T4 + T6)]				NS	NS	NS	NS	NS
Mo × Cu [(T2 + T4 + T7 + T9) vs. (T3 + T5 + T6 + T8)]				NS	NS	NS	NS	NS
Mo × S [(T2 + T3 + T8 + T9) vs. (T4 + T5 + T6 + T7)]				NS	NS	NS	NS	NS
Cu × S [(T3 + T4 + T7 + T8) vs. (T2 + T5 + T6 + T9)]				NS	NS	NS	NS	NS
Mo × Cu × S [(T3 + T4 + T6 + T9) vs. (T2 + T5 + T7 + T8)]				NS	NS	NS	NS	NS

*p*-value for treatment: 0.0001; *p*-value for time: 0.0001. Treatment × time: 0.0001. SEM—standard error of the mean; NS—not significant.

**Table 9 animals-13-02945-t009:** Serum ceruloplasmin activity (U/L) of lambs receiving a control diet or supplemented with molybdenum (Mo) or sources of copper (Cu) and sulfur (S).

Mineral Inclusion			Treatments	Time (Days)				
Mo	Cu	S		0	28	56	84	Mean
Without	Without	Without	T0	21.58	17.28	23.37	20.15	20.59
With	Without	Without	T1	13.32	14.82	17.06	11.44	14.16
Without	Inorganic	Inorganic	T2	22.96	17.12	17.98	15.65	18.3
		Organic	T3	20.7	15.85	18.26	21.41	19.05
	Organic	Inorganic	T4	18.85	15.99	13.02	11.43	14.82
		Organic	T5	17.33	15.76	16.62	16.88	16.65
With	Inorganic	Inorganic	T6	13.04	14.47	12.73	8.02	12.06
		Organic	T7	19.63	15.24	16.59	18.26	17.43
	Organic	Inorganic	T8	17.73	17.04	17.86	15.92	17.14
		Organic	T9	14.38	16.34	13.12	11.03	13.71
Principal effects								
Without					16.18	16.47	16.34	17.21
With					15.77	15.07	13.31	15.09
	Inorganic				15.67	16.39	15.84	16.71
	Organic				16.28	15.15	13.81	15.58
		inorganic			16.15	15.4	12.75	15.58
		Organic			15.8	16.14	16.89	16.71
Average data								
Mean				18.08	16.05	16.98	15.26	16.59
SEM				0.76	0.64	0.97	0.82	0.41
Statistical Probabilities of Contrasts								
Control vs. others [(T0) vs. (T1 + T2 + T3 + T4 + T5 + T6 + T7 + T8 + T9)]				0.0345	NS	NS	0.0013	0.0053
Mo vs. Others (T1 vs. T0 + T2 + T3 + T4 + T5 + T6 + T7 + T8 + T9)				0.043	NS	NS	NS	NS
Mo presence (T0 vs. T1)				0.0196	NS	NS	0.0254	NS
Cu source [(T3 + T4) vs. (T5 + T6)]				NS	NS	NS	NS	NS
S source [(T3 vs. T5) vs. (T4 + T6)]				NS	NS	NS	0.031	NS
Mo × Cu [(T2 + T4 + T7 + T9) vs. (T3 + T5 + T6 + T8)]				NS	NS	NS	NS	NS
Mo × S [(T2 + T3 + T8 + T9) vs. (T4 + T5 + T6 + T7)]				NS	NS	NS	NS	NS
Cu × S [(T3 + T4 + T7 + T8) vs. (T2 + T5 + T6 + T9)]				NS	NS	NS	0.0055	NS
Mo × Cu × S [(T3 + T4 + T6 + T9) vs. (T2 + T5 + T7 + T8)]				NS	NS	NS	0.0074	NS

*p*-value for treatment: 0.0246; *p*-value for time: 0.0021. Treatment × time: 0.0349. SEM—standard error of the mean; NS—not significant.

## Data Availability

The datasets generated and analyzed during the current study are available from the corresponding author upon reasonable request.

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
