# Peer review of "Influence of Molybdenum and Organic Sources of Copper and Sulfur on the Performance, Carcass Traits, Blood Mineral Concentration, and Ceruloplasmin Activity in Lambs"

_animals, 2023, doi:10.3390/ani13182945_

Round 1

Reviewer 1 Report

Dear authors,

The theme is relevant with interest, however, it is more difficult since there are already many works in this area, being necessary to innovate.

Attached I leave all the comments to the submitted manuscript to improve it.

The biggest doubt is the failure to present the ethical and animal welfare license to work with lambs in cages.

But I leave it to the editor's consideration.

Author Response

Reviewer 1

This paper aims to evaluate the addition of MO, together with Cu and S (organic vs inorganic).

A simple study, but with expected conclusions. However, the subject of the work is not new, as

there are already many works on this subject in small ruminants. The n in each group is too low

to be able to work the data statistically.

AU: Thank you for the considerations. Although this subject is not new, studies reporting the interaction among these 3 minerals, few studies, to our knowledge, reported the organic and inorganic interactions. So, this current study brings up new information on negative or positive organic and inorganic interactions of Cu and S in small ruminants. Regarding the n in each group, we recognize that is a huge limitation of this study; however, we respectfully ask the reviewer to consider our manuscript since this number of experimental unities has been found in the literature (Zhou J, Ren Y, Wen X, Yue S, Wang Z, Wang L, et al. Comparison of coated and uncoated trace elements on growth performance, apparent digestibility, intestinal development and microbial diversity in growing sheep. Frontiers in Microbiology. 2022;13 December:1–17). Besides the sentence justifying the n in each group in the M&M section, we added a sentence in the Discussion, explaining the limitation of this study due to the n in each group, as follows: “In addition, as a limitation of this study, the experimental unit in each group is low, thus there is a high chance of increased type II error. However, Zhou et al., (2022) in their actual study used similar design comparing coated and uncoated trace elements on growth performance, apparent digestibility, intestinal development and microbial diversity in growing sheep.”  

This work involves the supplementation of live animals, which are slaughtered at the end of the

trial, so this work should have an ethics and animal welfare authorisation according to the

policies and legislation for research. They only make reference to the slaughter rules, this is

insufficient for scientific work.

AU: This sentence: “The experimental procedures were reviewed and approved by the Bioethics Committee for Animal Experimentation of the College of Animal Science and Food Engineering (FZEA/USP; #14.1.1410.74.7).” has been added to the Institutional Review Board Statement of the manuscript, as recommended by the Animals MDPI Journal. However, we added it in the M&M section. Thank you.

Acronyms should only be used after they have been described and then the acronyms should

always be used. Throughout the text sometimes acronyms are used and sometimes the word is

writtien. You should correct this throughout the paper.

AU: Thank you. We have fixed it throughout the manuscript.

Abstract: The abstract clearly indicates the main objective of the work, although in the results

they refer to the quality of the carcasses and do not indicate this in the objectives, they should

do so. The experimental design is mentioned, but there is no reference to the methodologies

used. They refer to the results in a general way ending with a conclusion. The abstract is a bit

confusing; I suggest the authors rewrite the abstract in a clearer way.

AU: We added the carcass evaluations in the objective sentence of the abstract and in the introduction section. Regarding the methodology’s description, we did not mention them due to the number of words limitation. As in the Instruction for authors of the Animals journal, the abstract should be a total of about 200 words maximum. We have rewritten the abstract to make it clearer. We have added the results in a more detailed way.

Introduction:

The introduction is well written and easy to read. However, the references used are very old,

only 1 reference is recent (2022), only 3 references are from the last 10 years, that is, more

than 70% of the references used are more than 10 years old. There are numerous recent

papers, including in Animals where the physiological mechanisms of trace elements as well as

antagonism processes are addressed. Furthermore, the focus of the paper is on the origin of

trace elements, organic vs inorganic, and there is not a single reference to this topic in the

introduction. You should include a paragraph on this topic in your introduction.

AU: We added more recent references in the introduction. Also, we have added a paragraph describing the organic vs. inorganic trace element effects in ruminants. Thank you for the recommendation.

Material and Methods

Cages of only 1 m2? You should refer to the legislation starting that it is allowed to have animals

in only 1 m2.

AU: We have added the Bioethics Committee for Animal Experimentation approval information in the M&M section: as follows: The experimental procedures were reviewed and approved by the Bioethics Committee for Animal Experimentation of the College of Animal Science and Food Engineering (FZEA/USP; #14.1.1410.74.7).

Although they justify the low number of animals, I think it is not statistically feasible for this

sample.

AU: Regarding the n in each group, we recognize that is a huge limitation of this study; however, we respectfully ask the reviewer to consider our manuscript since this number of experimental unities has been found in the literature (Zhou J, Ren Y, Wen X, Yue S, Wang Z, Wang L, et al. Comparison of coated and uncoated trace elements on growth performance, apparent digestibility, intestinal development and microbial diversity in growing sheep. Frontiers in Microbiology. 2022;13 December:1–17). Besides the sentence justifying the n in each group in the M&M section, we added a sentence in the Discussion, explaining the limitation of this study due to the n in each group, as follows: “In addition, as a limitation of this study, the experimental unit in each group is low, thus there is a high chance of increased type II error. However, Zhou et al., (2022) in their actual study used similar design comparing coated and uncoated trace elements on growth performance, apparent digestibility, intestinal development and microbial diversity in growing sheep.”  

What is aADF and aNDF? The correct acronym is ADF and NDF.

AU: We have fixed it.

The units in Table 1 should be indicated in front of each parameter, not in the figure legend.

AU: The units were indicated in the title of the table. We performed a review through the recent papers from Animals and we identified that several studies presented their results as in the current study. We respectfully ask the reviewer to keep this table as it is, so we can decrease unnecessary information in it. Thank you.

Blood should have been taken at the beginning of the trial, day zero (0d), to be able to observe

the evolution of the supplementation effect.

AU: We, indeed, did this 0 d collection. In the M&M you can find the following sentence: For blood sampling handling, the animals were restrained in the pens to minimize movement and stress. Blood samples were collected at enrollment… The enrollment was the 0 d. However, we have added the expression “(0 d)” in the sentence to make it clearer. Thank you.

Results

Suggestion: You could present the nutritional values of the base diet in a table. it is easier to

read.

AU: We have presented the nutritional values of the basal diet in the table 1.

Why do tables 3 and 4 not present the statistical results for all the days under study?

AU: The statistical approach for days was presented in the table legend. Regarding the figures, the only day which interacted with treatment was the d 84. So, we have decided to present only this interaction to decrease unnecessary information in the manuscript.

What do the bars on the graphs represent? Are they the same in all graphs and are they

correct?

AU: The bars are the Standard Error of the Mean, presented as the pooled SEM. We have added more information in the legends. Thank you.

Discussion

The discussion is easy to read and manages to address and justify the results obtained in the

work. However, it presents the same problem as the introduction, the references used are not

recent.

AU: We have added some more recent citations. Thank you.

The bibliographical references throughout the discussion are incorrect, you should use the

reference as a number. For example, in line 297, you should write "Dick et al. (1975) [18]

reported...". There are several that only have the authors, but the reference number should be

included.

AU: Thank you. We have fixed it.

They should add at the end of the discussion, how this problem of trace element antagonism

can be overcome, suggesting for example the optimum point between these 3 minerals.

AU: We have added some sentences in the discussion section to make it clearer.

Conclusion- The last sentence suggests that there is no set standard for each variable studied.

You should not take the part for the whole. They should say that in this work, it was not

possible to identify a pa?ern in the variables studied, however further studies are needed to

confirm that organic sources of Cu and S interacted alone without a defined pa?ern.

AU: We have added a sentence in the conclusion to make it clearer.

Reviewer 2 Report

Dear authors,

it is an interesting and extensive study, but I lack a clear intention for what purpose it was done. Was it to emphasize the adverse effect of molybdenum and sulfur or to compare the absorption of organic and inorganic sources of copper and their better utilization and incorporation to blood and tissues?

I summarized all my recommendations in the following comments:

Abstract

The abstract is comprehensible, and it characterizes the most important findings, but I lack a conclusion, the combination of individual elements most suitable or worse for lambs and why?

Introduction

In this section, the authors could explain more about how molybdenum and sulfur are incorporated into soils or plants and why there may be an excess of them in feed and their toxic levels.

Materials and Methods

Is missing in group T3-T9 dosage per kg of dry matter. Can you describe the method, how were doses of Mo, S or Cu mixed into the feed in such small concentrations to ensure their uniform distribution in daily doses?

Results

Better explain the division of groups and values interpretation in tables 3-9. It is not clear from the results in the tables which supplemented group it is (T0 -T9?). The statistical interpretation is also not clear in tables 3-9. It would be better if the authors indicated the statistical significance directly in the columns for individual fattening periods.

Discussion

Line 297 and 298: Wrong and old citations.

Line 309: Wrong citation.

Line 322: Wrong citation.

Line 324: Wrong and old citations.

Line 346: Wrong and old citations.

Line 354: Wrong and old citation.

Conclusion

There is no statement or recommendation why organic or inorganic sources of Cu or S should be given or excluded from the nutrition of lambs during the fattening period.

Author Response

Reviewer 2

Dear authors,

it is an interesting and extensive study, but I lack a clear intention for what purpose it was done. Was it to emphasize the adverse effect of molybdenum and sulfur or to compare the absorption of organic and inorganic sources of copper and their better utilization and incorporation to blood and tissues?

AU: It was the purpose of the study. We have improved the objective sentence to make it clearer. Thank you.

I summarized all my recommendations in the following comments:

Abstract

The abstract is comprehensible, and it characterizes the most important findings, but I lack a conclusion, the combination of individual elements most suitable or worse for lambs and why?

AU: We have added a sentence as a conclusion in the abstract. Also, we have improved the conclusion section. “In this current study, it was not possible to identify a pattern in the variables studied, however further studies are needed to confirm that organic sources of Cu and S interacted alone without a defined pattern.”

Introduction

In this section, the authors could explain more about how molybdenum and sulfur are incorporated into soils or plants and why there may be an excess of them in feed and their toxic levels.

AU: We have improved some sentences in the introduction section, emphasizing the organic and organic sources of Cu and S.

Materials and Methods

Is missing in group T3-T9 dosage per kg of dry matter. Can you describe the method, how were doses of Mo, S or Cu mixed into the feed in such small concentrations to ensure their uniform distribution in daily doses?

AU: We have improved the treatment description to make it clearer.

Results

Better explain the division of groups and values interpretation in tables 3-9. It is not clear from the results in the tables which supplemented group it is (T0 -T9?). The statistical interpretation is also not clear in tables 3-9. It would be better if the authors indicated the statistical significance directly in the columns for individual fattening periods.

Au: We have added more information in the tables to make it clearer.

Discussion

AU: We have fixed all the citation in the manuscript and added some more recent references to explain our results.

Line 297 and 298: Wrong and old citations.

Line 309: Wrong citation.

Line 322: Wrong citation.

Line 324: Wrong and old citations.

Line 346: Wrong and old citations.

Line 354: Wrong and old citation.

Conclusion

There is no statement or recommendation why organic or inorganic sources of Cu or S should be given or excluded from the nutrition of lambs during the fattening period.

AU: we have added a sentence in the conclusion explaining our study. Thank you.

Reviewer 3 Report

The manuscript studies the Influence of molybdenum and organic sources of copper and sulfur on performance and blood mineral concentration in lambs.

However, the following remarks must be made.

1. When studying the inclusion of minerals in the diet, why is the bioavailability of these elements not studied? They are of different origin and nature, so they will have different bioavailability? Why hasn't it been studied? How can it be argued that these elements were assimilated in the body of animals and got into metabolic processes for a long time!

2. The abstract of the manuscript should be written more clearly. Specify how the experimental groups T1-T9 were supplemented with minerals. Also, when describing, indicate how much the established effect differed in the groups relative to the control.

3. Make corrections either in the purpose of the study or in the title of the manuscript. The title does not mention the study of the activity of ceruloplasmin.

4. When describing the basic diet of animals, there is no description of the nature of the constituent minerals. It is very important! It is also not clear what were the amounts of the studied minerals in animal groups relative to the recommended norms!

5. The conclusion needs to be redone. It does not evaluate the results obtained with their analysis!

A file with my comments and recommendations on the manuscript is attached to the review!

Eliminating these comments will make the manuscript better and more understandable!

The manuscript has minor errors and typos.

Therefore, a minor revision of the English in this manuscript is required!

Author Response

Reviewer 3

  1. When studying the inclusion of minerals in the diet, why is the bioavailability of these elements not studied?They are of different origin and nature, so they will have different bioavailability?Why hasn't it been studied? How can it be argued that these elements were assimilated in the body of animals and got into metabolic processes for a long time!

AU: Thank you for your consideration. In this study, we did not evaluate bioavailability since it was not our goal. And yes, we agree with all your claims, but we do not have outcomes to argue on the assimilation of these minerals. We agree that it is a weakness of our study, but we ask respectfully to the reviewer consider this manuscript for publication. 

  1. The abstract of the manuscript should be written more clearly. Specify how the experimental groups T1-T9 were supplemented with minerals. Also, when describing, indicate how much the established effect differed in the groups relative to the control.

AU: We have rewritten the abstract to make it clearer.

  1. Make corrections either in the purpose of the study or in the title of the manuscript. The title does not mention the study of the activity of ceruloplasmin.

AU: We have improved the title to make it clearer.

  1. When describing the basic diet of animals, there is no description of the nature of the constituent minerals. It is very important! It is also not clear what were the amounts of the studied minerals in animal groups relative to the recommended norms!

AU: We have improved the M&M description to make it clearer.

  1. The conclusion needs to be redone. It does not evaluate the results obtained with their analysis!

AU: We have rewritten the conclusion to make it clearer. We really appreciate all your comments.

A file with my comments and recommendations on the manuscript is attached to the review!

Eliminating these comments will make the manuscript better and more understandable!

Round 2

Reviewer 1 Report

Dear authors,

The article has been greatly improved by the changes that have been made.
My doubt, although they have justified it properly, remains in the number of animals used. Although it has already been done by other authors, it does not mean that one can accept these data. In my opinion, despite not having access to the raw data, the statistics applied to some data are not correct. Given my experience, I think that the data do not show a normal distribution and the data should be worked with non-parametric data. However, as I have already mentioned I do not have access to the data, so I leave it to the editor to request this data.
In any case, the changes introduced have benefited the article.

Author Response

Reviewer 1 – Round 2

The article has been greatly improved by the changes that have been made.
AU: Thank you very much for all the help. We really appreciate it.

My doubt, although they have justified it properly, remains in the number of animals used. Although it has already been done by other authors, it does not mean that one can accept these data. In my opinion, despite not having access to the raw data, the statistics applied to some data are not correct. Given my experience, I think that the data do not show a normal distribution and the data should be worked with non-parametric data. However, as I have already mentioned I do not have access to the data, so I leave it to the editor to request this data.

AU: As stated in the manuscript (M&M section) “The normality of the residuals was verified by the Shapiro-Wilk test using the UNIVARIATE procedure”. Thus, we did not detect any non-parametric data.  If so, we will describe it in the manuscript running a different statistical approach. We really thank all the concern addressed regarding the low experimental unit herein used. Based on our experience, this type of study involving three-ways interactions is of huge importance to understand the effects of several minerals in one unique study; however, we need to increase our facilities to fit more powerful statistical analysis to perform it. To be clear with the readers we have added a sentence in the first paragraph of the discussion section explaining the study limitations regarding the low experimental unit enrolled. Finally, we understand the concerns on this topic in our study, but we ask respectfully to the reviewer to accept our trial since it brings up important findings is this study field, although some limitations exist.

In any case, the changes introduced have benefited the article.

AU: Thank you

Reviewer 2 Report

I thank the authors for taking note of all my comments and incorporating them into the manuscript.

Author Response

Reviewer 2 – Round 2

I thank the authors for taking note of all my comments and incorporating them into the manuscript.

AU: we really appreciate all the concerns addressed to us, it was important to make the manuscript better.
